# Discriminating Bacterial Infection from Other Causes of Fever Using Body Temperature Entropy Analysis

**DOI:** 10.3390/e24040510

**Published:** 2022-04-05

**Authors:** Borja Vargas, David Cuesta-Frau, Paula González-López, María-José Fernández-Cotarelo, Óscar Vázquez-Gómez, Ana Colás, Manuel Varela

**Affiliations:** 1Department of Internal Medicine, Hospital Universitario de Móstoles, 28935 Mostoles, Spain; paula.gonzalez.lopez@salud.madrid.org (P.G.-L.); mfcotarelo@salud.madrid.org (M.-J.F.-C.); ovazquez@salud.madrid.org (Ó.V.-G.); manuel.varela@salud.madrid.org (M.V.); 2Technological Institute of Informatics, Universitat Politècnica de València, Alcoi Campus, 03801 Alcoi, Spain; dcuesta@disca.upv.es; 3Faculty of Health Sciences, Universidad Rey Juan Carlos, 28922 Alcorcon, Spain; 4Department of Internal Medicine, Hospital Universitario 12 de Octubre, 28041 Madrid, Spain; anamaria.colas@salud.madrid.org

**Keywords:** time series, body temperature, Slope Entropy, Approximate Entropy, Sample Entropy, classification, fever

## Abstract

Body temperature is usually employed in clinical practice by strict binary thresholding, aiming to classify patients as having fever or not. In the last years, other approaches based on the continuous analysis of body temperature time series have emerged. These are not only based on absolute thresholds but also on patterns and temporal dynamics of these time series, thus providing promising tools for early diagnosis. The present study applies three time series entropy calculation methods (Slope Entropy, Approximate Entropy, and Sample Entropy) to body temperature records of patients with bacterial infections and other causes of fever in search of possible differences that could be exploited for automatic classification. In the comparative analysis, Slope Entropy proved to be a stable and robust method that could bring higher sensitivity to the realm of entropy tools applied in this context of clinical thermometry. This method was able to find statistically significant differences between the two classes analyzed in all experiments, with sensitivity and specificity above 70% in most cases.

## 1. Introduction

Body temperature is a key clinical parameter. It is usually assessed once per shift in hospital wards and has always been considered a hallmark of infectious diseases. However, the values obtained with the standard measurements are interpreted dichotomously: either the patient has a fever or is afebrile.

Body temperature assessment is also highly dependent on the method of measurement. Central methods are accurate and reliable (pulmonary artery catheter, urinary bladder, esophagus) but they are not suitable in most clinical scenarios. Tympanic temperature is often used as a replacement for central temperature because values are close, and it is more convenient and less invasive [1]. Peripheral temperature can be assessed in different anatomical locations (mouth, armpit). Despite not being as accurate [2], peripheral measurements are the standard procedure in clinical practice.

Furthermore, the definition of fever is arguably flawed, as it depends on many factors such as age, gender, circadian rhythms, or underlying conditions [3,4,5]. As a matter of fact, there is no universal threshold for fever, as a wide range of temperatures has been shown in individuals considered healthy [6,7]. Some efforts to standardize the *normal* body temperature range have been carried out in the past [7] but they have not been transferred into clinical practice.

Traditionally, attempts have been made to find clinical differences in the patterns of fever caused by infectious diseases (malaria, tuberculosis, typhoid fever) [8]. Nevertheless, none of these approaches are sufficient to make clinical decisions [8,9]. Furthermore, a wide spectrum of noninfectious conditions can also induce the synthesis and release of pyrogenic cytokines and eventually cause fever [10].

Since body temperature regulation is a dynamical process, by obtaining just two or three measurements per day, a wealth of information is lost. However, some devices are available to obtain high-frequency measurements of body temperature. Body temperature monitoring has been proven useful in certain clinical scenarios when more frequent measurements (associated or not to alarm settings) entail earlier recognition of fever [11,12,13,14]. Moreover, temperature monitoring devices enable the registry of temperature time series. This allows its use as a continuous variable, instead of a series of isolated values [15].

Similar to many other biological systems, thermoregulation can be considered a complex process, and might therefore be analyzed under the scope of nonlinear dynamics. Complexity metrics have previously been applied to other biological variables [16]. It has been widely demonstrated that changes in complexity of biological signals are associated with damage to or degradation of the system [17,18,19,20,21,22,23].

In this context, entropy statistics could be of clear interest to unveil certain characteristics of the thermoregulation process and, perhaps, the underlying cause of fever. In previous works, we have already demonstrated the feasibility of this approach. For example, in [24], we described a method based on the entropy statistic Slope Entropy (SlpEn) [25] to distinguish between body temperature time series from malaria and dengue patients. The achieved accuracy was up to 90% correctly classified records with a single numerical feature computed for each one. In another study [26], a different entropy method, Sample Entropy (SampEn) [27], was used with the same purpose of distinguishing among body temperature time series coming from infectious diseases, tuberculosis, nontuberculosis, and dengue fever patients. The global accuracy achieved was close to 70%.

Other works have used a combination of features; this is the case for the work described in [28], which used temperature temporal patterns to detect tuberculosis. In [29], the authors used a more sophisticated approach using the Fourier transform, entropy, energy, power, and a set of additional coefficients to train a quadratic support vector machine to carry out the classification of tuberculosis, intracellular bacterial infections, dengue, and inflammatory and neoplastic diseases temperature time series.

The goal of this study is to assess if patients with bacterial infections have significant changes in the entropy of their body temperature compared with patients with other infections or other causes of fever. As the entropy statistic for the analysis, we chose SlpEn for its good performance in previous studies [24,30,31]. For comparative purposes, we included more widely used methods such as Approximate Entropy (ApEn) [32] and SampEn, which have been successfully used in a myriad of similar biosignal classification works [33,34,35,36,37,38,39].

## 2. Materials and Methods

### 2.1. SlpEn

The recently proposed time series entropy measure termed Slope Entropy (SlpEn) [25] can achieve high classification accuracy using a diverse set of records [24,25,30]. Despite its short life, it has already been implemented in scientific software tools such as EntropyHub (https://github.com/MattWillFlood/EntropyHub.jl, accessed on 15 February 2022) and CEPS (Complexity and Entropy in Physiological Signals) [40].

The first step of SlpEn computation is extraction from an input time series $x=\left\{x_{0},x_{1},\ldots ,x_{N-1}\right\}$ of a set of consecutive overlapping subsequences of length *m*, commencing at sample *i*, $x_{i}=\left\{x_{i},x_{i+1},\ldots ,x_{i+m-1}\right\}$, 0≤i<N−m+1 (*m* being the embedded dimension variable and *n* the total length of the time series, with m<<N). Each of the $n_{m}$ extracted subsequences, $x_{i}$, can then be transformed into a new one of length m−1 by computing and storing the differences between each pair of consecutive samples in the subsequence, namely, $y_{i}=\left\{x_{i}-x_{i+1},x_{i+1}-x_{i+2},\ldots ,x_{i+m-2}-x_{i+m-1}\right\}$.

Using, in its basic configuration [25], 5 different symbols from an alphabet—for example +2, +1, 0, −1, −2—the differences obtained are represented by these symbols instead, according to two input thresholds, δ and γ, and the expressions described in [25]. Further details of SlpEn implementation and examples can be found in [24,25]. A software library using this method is also described in [40].

In addition to SlpEn, two other entropy methods—ApEn and SampEn—were applied to the time series in order to assess the hypothesized improvement in classification accuracy that SlpEn could bring to the analysis. Although these methods have been used extensively, and they are characterized and described in great detail in a number of publications [41,42,43,44,45,46], they are depicted for completeness in the next two subsections.

### 2.2. Approximate Entropy

ApEn [32] is also based on extracting subsequences of length *m* from the input time series, $x_{i}=\left\{x_{i},x_{i+1},\ldots ,x_{i+m-1}\right\}$, as for SlpEn. Then, a distance is computed between every subsequence and a fixed reference $x_{j}$, $d_{ij}=\max(|x_{i+k}-x_{j+k}\left|\right),$ with 0≤k≤m−1.

If the number of comparisons falling below a predefined threshold *r*—termed matches, $d_{ij}<r$—is computed for two consecutive embedded dimensions (*m* and m←m+1), two counters can be defined as $B_{i}\left(r\right)$, number of *j* so that $d_{i,j}^{m}\leq r$, and $A_{i}\left(r\right)$, number of *j* so that $d_{i,j}^{m+1}\leq r$, with 0≤j<N−m+1.

Computing the averages of these counters, $B_{i}^{m}\left(r\right)=\frac{1}{N-m+1}B_{i}\left(r\right)$ and $A_{i}^{m}\left(r\right)=\frac{1}{N-m}A_{i}\left(r\right)$, the main ApEn variables are calculated as $\phi^{m}\left(r\right)=\frac{1}{N-m+1}\sum_{i=1}^{N-m+1}\log B_{i}^{m}\left(r\right)$ and $\phi^{m+1}\left(r\right)=\frac{1}{N-m}\sum_{i=1}^{N-m}\log A_{i}^{m}\left(r\right)$, from which the result of ApEn can be finally obtained as $\mathrm{ApEn}\left(m,r,N\right)=\left[\phi^{m}\left(r\right)-\phi^{m+1}\left(r\right)\right]$.

### 2.3. Sample Entropy

The first steps of the SampEn algorithm [35] are the same as for ApEn. However, when counting the matches, subsequences are not compared with themselves, formally 0≤j<N−m+1, with j≠i.

Then, the statistics are now $B^{m}\left(r\right)=\frac{1}{N-m}\sum_{i=1}^{N-m}B_{i}^{m}\left(r\right)$ and $A^{m}\left(r\right)=\frac{1}{N-m}\sum_{i=1}^{N-m}A_{i}^{m}\left(r\right)$, from which SampEn is computed as $\mathrm{SampEn}\left(m,r,N\right)=-\log\left[\frac{A^{m}\left(r\right)}{B^{m}\left(r\right)}\right]$.

### 2.4. Experimental Dataset

The study was conducted at Hospital Universitario de Móstoles (Madrid, Spain). Patients older than 18 years old admitted to the general Internal Medicine ward presenting with fever at admission and/or suspected infectious disease were considered suitable for inclusion. Pregnancy and inability to cooperate with the monitoring process were considered exclusion criteria.

Temperature values were obtained through a probe (Truer Medical, Inc., Orange, CA, USA) placed in the external auditory canal (EAC), after otoscopy to check the integrity of the tympanic membrane. Data from the EAC were used as surrogates of central temperature [1]. The probe was wired to a Holter device (TherCom, Innovatec) that registered one measurement per minute. When feasible, the monitoring process was performed in real-time. Otherwise, data were stored in the device and downloaded later for analysis. The aim was to perform 24-h recordings, but in some cases, the process was stopped earlier due to poor compliance of the patient, displacement of the probes for long periods (preventing the proper recording of data), or abnormally low temperatures, suggesting that measurements were clearly inaccurate.

Patients were classified into two categories concerning diagnosis: bacterial infection (confirmed or suspected) or others. The latter included patients with nonbacterial infections (viral, fungal, etc.) or with fever caused by inflammatory diseases, cancer, or fever of unknown cause (when bacterial infection was deemed to be excluded).

Temperature time series were processed by visual inspection. In some cases, the beginning and/or the end of the recording were trimmed to ensure the stability of the signal. Disconnections of at most 5 measurements were repaired through linear interpolation. For longer disconnections, the segment was removed, provided the remaining interval was clean.

The experimental dataset contained 10 body temperature time series of patients with confirmed bacterial infection and 13 from patients with other causes of fever. The lengths of the time series are shown in Table 1. In all cases, different fixed time series lengths were used to assess the classification accuracy of each metric and ensure length equality: 500, 600, 700, 800, 900, 1000, and 1100. Those time series below the cut-off length were discarded in that specific experiment.

Figure 1 depicts two body temperature records from this database, with one from each class (fever caused by a bacterial infection, and fever caused by a different clinical condition).

## 3. Experiments and Results

All the experimental time series were processed using the three entropy calculation methods described previously: SlpEn, ApEn, and SampEn. They were also cut short to the lengths stated above. The remaining central part was used for the analysis, as it would theoretically be the most stable segment (thermal equilibrium reached, probe still in place). The entropy result was used as the classification feature applying Sensitivity (Se) and Specificity (Sp) [47], with a threshold obtained from the corresponding ROC curve (closest point to (0,1); an example is shown in Figure 2) [48,49,50]. The statistical significance was assessed using the Wilcoxon–Mann–Whitney test [51], with α=0.05. Input parameters were varied in the range m∈[3,9] and γ,r∈[0.10,0.90]. For SlpEn, δ was kept constant at δ=0.001. Time series were normalized for zero mean and unit standard deviation. The stationarity of the input time series was assessed by computing the standard deviation for each consecutive 50-sample window, yielding fairly similar values.

Table 2 shows the results for lengths N=500,600, and 700. For N=500, SlpEn achieved good classification accuracy for m=3 and in the γ region of 0.15–0.30. Neither ApEn nor SampEn reached significance for any combination of their input parameters *m* and *r*, after a grid search for *m* between 3 and 9, and *r* between 0.10 and 0.90 in 0.05 steps. It is important to note that these methods are very sensitive to length, and N=500 is arguably too short for them.

For N=600 and N=700 the results were similar. ApEn and SampEn did not discriminate, SlpEn remained stable in the same region of m=3 and γ= 0.15–0.25, but other parameter configurations in the same region of γ with values of *m* such as 4, 6, and 7 also reached discriminatory power.

Table 3 displays the results for lengths N=800 and 900. For N=800, the trend is the same as in Table 2. SlpEn is able to find differences in the vicinity of m=3 and γ=0.20, but ApEn and SampEn are unable to yield any statistically significant classification. For N=900, the number of parameter combinations for SlpEn increases, with γ fairly stable in the same region around 0.20, for almost any *m* value except 5. Additionally, ApEn is also significant in the region around m=3 and r=0.20.

Finally, the results for N=1000 and N=1100 are shown in Table 4. No more lengths were tested since not enough time series would be available if N>1100. For both cases, the number of significant combinations increased significantly, with SlpEn certainly stable in the same regions as for other *N* values, and even ApEn and SampEn reaching significance for N=1100. Figure 3 shows a plot of results for N=1100,γ=r=0.20,m=3 and for the three methods tested.

## 4. Discussion

The experiments explored the capability of SlpEn, ApEn and SampEn to distinguish between two classes of body temperature records: time series from patients with bacterial infection and time series from patients also with fever but due to other causes. The experimental set available enabled a study using lengths from 500 up to 1100 samples.

For all these lengths, SlpEn was able to find significant differences when the input parameters were m=3 and γ= 0.15–0.25, with additional *m* values available depending on *N* values. This illustrates the fact that SlpEn is fairly stable and robust, as also demonstrated in other studies based on this recent method [30,31,52].

The results obtained using ApEn were significant only for N=900 (Table 3) and N=1100 (Table 4). This is in accordance with the reported high sensitivity of ApEn to the length of the input time series [44]. One of the most popular guidelines for this minimum length using ApEn is $N\geq 10^{m}$ [53], which translates in this case to N≥1000, in agreement with the results in this study. In order to illustrate how the ApEn statistics were computed for the most unfavorable case, N=500, the percentage of estimated probabilities on at least 10 matches, weak criterion, was 80.17±10.37, and on at least 100 matches, strong criterion, was 6.52±13.01 [54].

SampEn only achieved significance for N=1100. Although SampEn is known to be sensitive to input time series length, it is usually claimed to be more robust in this regard than ApEn. However, in other cases, we have also found that ApEn performed better than SampEn in classification tasks such as in [38].

There is also an association between time series length and classification performance in terms of specificity and sensitivity [47]. For shorter time series, there are some cases where these metrics achieve values below 0.7. As the length increases, the performance improves, with more values in the vicinity of 0.85, arguably very high for a biomedical signal classification application.

Therefore, for lengths shorter than 1000 samples, imposed by the operational difficulties linked to obtaining long-term body temperature records, SlpEn seems a good choice to find differences between the record classes present in the experimental dataset. If longer series were available, other methods such as ApEn and SampEn could also be applied, which remains to be further studied.

From a clinical perspective, the results of this work suggest that patients admitted to the hospital with a diagnosis of bacterial infection had a misregulation of their body temperature, measured with entropy statistics. This is in accordance with findings of a previous study by our group [55]. The complexity of biosignal time series seems to be an indicator of the integrity and performance of biological systems, and disease usually exhibits low levels of entropy metrics [16,20]. Bacterial infection may entail the development of sepsis, a clinical entity with a very high risk of complications and death, which is unusual with other infections or other causes of fever. In our opinion, the loss of entropy that we have observed in the temperature curves of patients with bacterial infections may be another facet of homeostasis disturbance.

Remarkably, these results were irrespective of the confirmation of fever by the staff nurse (standard measurements), or the maximum temperature obtained. These results suggest that body temperature may supply relevant information, over and above attaining a certain pre-established *febrile threshold*.

In fact, body temperature may provide clues in many clinical aspects, as long as enough information is obtained through continuous monitoring. It has already proved to be useful to assess the prognosis of critically ill patients in the Intensive Care Unit [19,56], to forecast fever peaks [13], and to classify patients according to the cause of fever [26,29].

We are aware of the many limitations of this work. On the one hand, the monitoring system has some issues: the tympanic probe is prone to be displaced, wired probes can be bothersome for the patient, the Holter device needs to be wirelessly connected to a computer in the range of Bluetooth, etc. For these reasons, over half of the patients in the study were excluded from this analysis because the recordings were lost or defective. Several adjustments were carried out during the study to solve or reduce the impact of these issues, such as real-time monitoring through a wireless network and periodic backups of data to keep a copy of the recording in case there was a disconnection. In any case, we acknowledge that the final sample was small and, although significant differences have been found between the two groups, reliability might be limited for this reason.

On the other hand, as it has already been exposed, entropy metrics need clean time series and are in general more informative the longer the data. This has been a limitation in this work and is a common problem for the analysis of biological time series recorded in real life, as many factors may cause artifacts and make data unsuitable for evaluation.

In our opinion, future research should focus on two issues: Firstly, acquisition of long and clean time series. For this purpose, wireless and ergonomic probes that fit properly at the external auditory canal could improve the quality of the recordings and minimize the loss of information. Secondly, obtaining temperature recordings in a wide range of clinical settings—including healthy individuals for comparison—may provide details about physiological processes and may broaden the utility of clinical thermometry to subtler issues than just the identification of fever peaks.

## Figures and Tables

**Figure 1 entropy-24-00510-f001:**
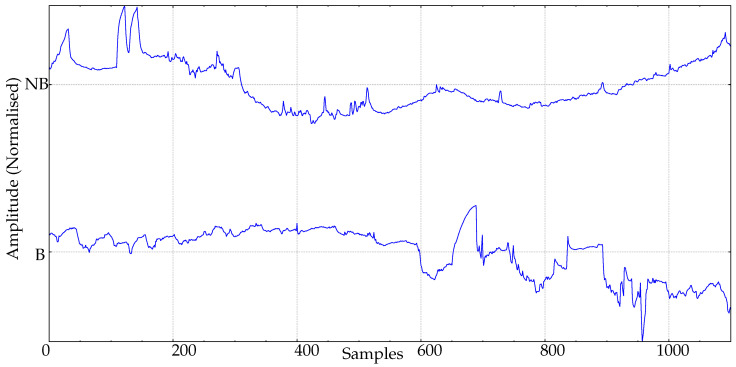
Example of body temperature records from the experimental database. B: Bacterial infection. NB: Nonbacterial cause of fever.

**Figure 2 entropy-24-00510-f002:**
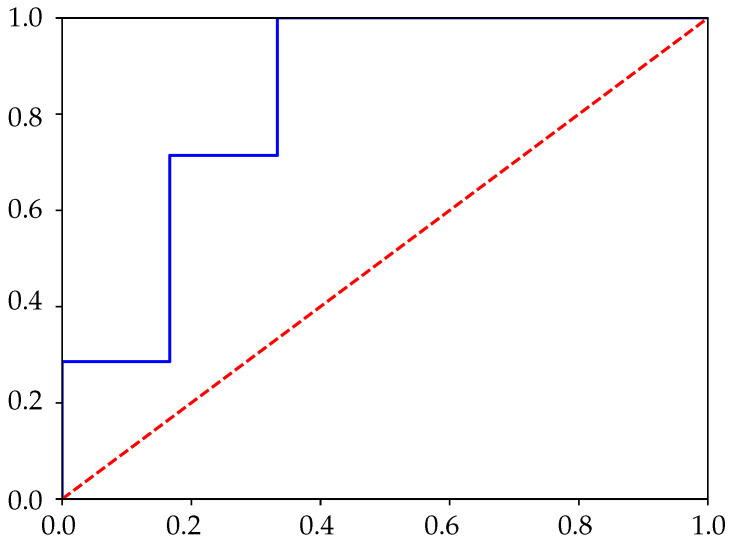
Example of ROC curve using Slope Entropy (SlpEn), m=3, and γ=0.20.

**Figure 3 entropy-24-00510-f003:**
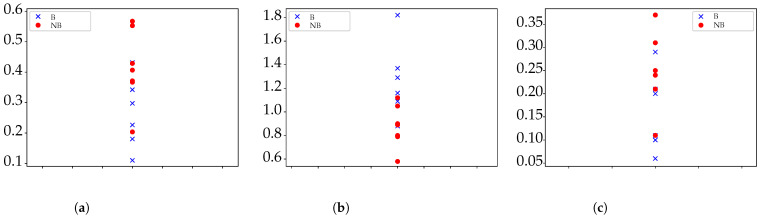
Example of graphical results for each method tested with N=1100. SlpEn results have been inverted and rescaled for better visualization. (**a**) Results for ApEn with r=0.2 and m=3. (**b**) Results for SlpEn with γ=0.2 and m=3 (In absolute value and normalized by 100). (**c**) Results for SampEn with r=0.2 and m=3.

**Table 1 entropy-24-00510-t001:** Original lengths of the body time series used in the experiments. Length is defined in terms of number of samples, taking into account that the sampling frequency was one sample per minute.

	1	2	3	4	5	6	7	8	9	10	11	12	13
Bacterial infection	936	1231	1154	1279	1443	1134	680	710	586	1117	–	–	–
Other causes of fever	1284	1468	1427	1444	1295	913	1105	1017	830	537	1121	859	934

**Table 2 entropy-24-00510-t002:** Experiment results for lengths N=500,600,700 using SlpEn, Approximate Entropy (ApEn), and Sample Entropy (SampEn). Parameter grid search for *m*, between 3 and 9, and *r* and γ, between 0.10 and 0.90 in 0.05 steps. The values of the input parameters are included as (m,r) or (m,γ) for cases when p<0.05 after the grid search. Otherwise, no combination provided significant results, represented by −−. Statistical significance was only reached by SlpEn.

	N=500				N=600				N=700			
	**Parameters**	p	**Se**	**Sp**	**Parameters**	p	**Se**	**Sp**	**Parameters**	p	**Se**	**Sp**
SlpEn	(m=3, γ=0.15)	0.0255	0.90	0.76	(m=3, γ=0.15)	0.0190	0.88	0.75	(m=3, γ=0.15)	0.0448	0.87	0.75
	(3,0.20)	0.0407	0.69	0.61	(3,0.20)	0.0330	0.77	0.75	(3,0.20)	0.0448	1	0.66
	(3,0.25)	0.0299	0.80	0.76	(3,0.25)	0.0466	1	1	(4,0.15)	0.0448	0.87	0.66
	(3,0.30)	0.0545	1	0.61	(4,0.15)	0.0330	1	0.66	(6,0.10)	0.0307	0.66	1
	−−	p>0.05	−−	−−	(4,0.20)	0.0466	0.77	0.66	(7,0.20)	0.0372	0.75	0.87
ApEn	−−	p>0.05	−−	−−	−−	p>0.05	−−	−−	−−	p>0.05	−−	−−
SampEn	−−	p>0.05	−−	−−	−−	p>0.05	−−	−−	−−	p>0.05	−−	−−

**Table 3 entropy-24-00510-t003:** Experiment results for lengths N=800,900 using SlpEn, ApEn, and SampEn. Parameter grid search for *m*, between 3 and 9, and *r* and γ, between 0.10 and 0.90 in 0.05 steps. The values of the input parameters are included as (m,r) or (m,γ) for cases when p<0.05 after the grid search. Otherwise, no combination provided significant results, represented by −−. Statistical significance was reached by SlpEn and ApEn.

	N=800				N=900			
	**Parameters**	p	**Se**	**Sp**	**Parameters**	p	**Se**	**Sp**
SlpEn	(m=3,γ=0.15)	0.0179	1	0.75	(m=3,γ=0.10)	0.0247	0.71	0.80
	(3,0.20)	0.0425	0.85	0.66	(3,0.15)	0.0191	0.71	0.90
	(4,0.15)	0.0346	1	0.66	(3,0.20)	0.0317	0.71	0.90
	−−	p>0.05	−−	−−	(3,0.25)	0.0404	0.71	0.90
	−−	p>0.05	−−	−−	(4,0.10)	0.0146	0.71	0.90
	−−	p>0.05	−−	−−	(4,0.15)	0.0247	0.71	0.90
	−−	p>0.05	−−	−−	(4,0.20)	0.0317	0.71	0.90
	−−	p>0.05	−−	−−	(4,0.35)	0.0404	0.71	0.70
	−−	p>0.05	−−	−−	(6,0.10)	0.0191	0.80	0.85
	−−	p>0.05	−−	−−	(7,0.10)	0.0247	0.70	0.85
	−−	p>0.05	−−	−−	(7,0.15)	0.0317	0.70	0.85
	−−	p>0.05	−−	−−	(7,0.20)	0.0404	0.70	0.85
	−−	p>0.05	−−	−−	(8,0.10)	0.0317	0.80	0.85
	−−	p>0.05	−−	−−	(8,0.15)	0.0191	0.90	0.71
	−−	p>0.05	−−	−−	(8,0.20)	0.0317	0.90	0.71
	−−	p>0.05	−−	−−	(9,0.15)	0.0317	0.70	0.85
ApEn	−−	p>0.05	−−	−−	(m=3,r=0.20)	0.0317	0.85	0.7
	−−	p>0.05	−−	−−	(3,0.25)	0.0317	0.85	0.7
	−−	p>0.05	−−	−−	(4,0.20)	0.0404	0.85	0.7
	−−	p>0.05	−−	−−	(4,0.25)	0.0247	1	0.7
SampEn	−−	p>0.05	−−	−−	−−	p>0.05	−−	−−

**Table 4 entropy-24-00510-t004:** Experiment results for lengths N=1000,1100 using SlpEn, ApEn, and SampEn. Parameter grid search for *m*, between 3 and 9, and *r* and γ, between 0.10 and 0.90 in 0.05 steps. The values of the input parameters are included as (m,r) or (m,γ) for cases when p<0.05 after the grid search. Otherwise, no combination provided significant results, represented by −−. Statistical significance was reached by all methods in some cases.

	N=1000				N=1100			
	**Parameters**	p	**Se**	**Sp**	**Parameters**	p	**Se**	**Sp**
SlpEn	(m=3,γ=0.1)	0.0388	0.83	0.87	(m=3,γ=0.1)	0.0222	0.83	0.85
	(3,0.15)	0.0388	0.83	0.87	(3,0.15)	0.0222	0.83	0.85
	(4,0.10)	0.0388	0.83	0.75	(3,0.20)	0.0222	0.83	0.85
	(4,0.15)	0.0281	1	0.75	(3,0.25)	0.0151	0.83	0.85
	(8,0.10)	0.0388	0.75	0.83	(3,0.30)	0.0222	0.83	0.85
	(8,0.15)	0.0281	0.75	1	(3,0.35)	0.0101	1	0.85
	(9,0.10)	0.0388	0.75	0.83	(3,0.40)	0.0321	0.83	0.71
	(9,0.15)	0.0098	0.87	0.83	(4, 0.10–0.40)	0.0101	0.83	0.85
	(9,0.20)	0.0388	0.87	0.66	(6,0.10)	0.0222	0.85	0.83
	(9,0.25)	0.0281	0.75	0.83	(6,0.15)	0.0222	0.85	1
	(9,0.35)	0.0388	0.87	0.83	(7,0.10)	0.0151	0.85	1
	−−		−−	−−	(7,0.15)	0.0222	0.85	0.83
	−−		−−	−−	(8, 0.10–0.35)	0.0321	0.85	0.83
	−−		−−	−−	(9, 0.15–0.35)	0.0151	0.71	1
ApEn	−−		−−	−−	(m=9,r=0.1)	0.0151	0.85	0.83
	−−		−−	−−	(10,0.1)	0.0321	0.85	0.83
SampEn	−−		−−	−−	(m=3,r= 0.1–0.6)	0.0455	0.83	0.85

## Data Availability

The data presented in this study are available on request from the corresponding author. The data are not publicly available due to confidentiality reasons.

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
