# Peer review of "Discriminating Bacterial Infection from Other Causes of Fever Using Body Temperature Entropy Analysis"

_entropy, 2022, doi:10.3390/e24040510_

Round 1
Reviewer 1 Report
This study presents the application of slope entropy, approximate entropy, and sample entropy to body temperature records as a distinguishing feature between non–bacterial and bacterial febrile subjects.
The most important drawbacks are:
- In the section Materials and methods, there should be a brief description of all the methods that were used in the study. Please describe Approximate Entropy and Sample Entropy.
- In section 2.2. stationarity of data should be commented upon. Although, stationarity is not set as a requirement for SampEn, ApEn, it is necessary for reliable estimation of statistical moments such as SD... and the value of r is dependent on SD.
- It should be emphasized that this analysis was limited to a small dataset (only 10 body temperature time series of patients with confirmed bacterial infection, and 13 of other fever causes).
- Table 1 should include measurement units (samples or seconds).
- Figure 1 Missing units in the y-axis
- Ln 144-145: "not reached significance for any combination of their input parameters" is not precise enough. In Table 2, it should be written the value of paramers m and r for ApEn, and SampEn.
- Ln 175-176 failed to include a reference for recommendation parameters.
- It would be good to strengthen the study by checking the reliability of the estimated probabilities in ApEn and SampEn on short time series. In the literature, there are similar findings for cross-approximate entropy in cardiovascular and artificial environments. The estimated probability didn't satisfy even weak criteria for realiable estimation.
- Please revise the manuscript for grammar and spelling mistakes.
Author Response
We would like to thank the reviewer for the comments provided since they have clearly helped to improve the quality and comprehension of the text.
Please see the attachment with our point-by-point response.

Reviewer 2 Report
This is a very interesting paper. It has a simple goal, to investigate the possibility of distinguishing between bacterial and viral infections based on time series that contain temperature samples, sampled at a rate of once per minute and analyzed using three types of entropy. The authors did not fall into the trap of applying entropic formulas without thinking. They considered the limitations of the application of entropy, e.g. influence of the length of the time series. It is especially very good to normalize and centralize time series (although with regard to ApEn and SampEn the results would remain the same, regardless of whether the series are normalized and centralized or not).
Minor comments:
In order to perform normalization, the initial standard deviation must be estimated from each time series. The estimation of moments is justified only for stationary time series. Have you checked the stationarity? (My experience with temperature monitoring in Wistar rats is that the time series of temperatures were stationary).
It is not clear what Figure 1 represents: time series of temperatures or their normalized counterparts. Namely, units on y-axes are missing. The x-axis units are missing as well, but at least the number exists, and it can be presumed that it is in seconds. On the other hand, the units shown on the y-axis are important as they can also show whether temperature fluctuations are very pronounced or the axis scale is just increased (it is important for stationary).
It is common to include a control group of healthy volunteers in such experiments. SlpEn showed a statistically significant difference between viral and bacterial infection, but would these two groups differ from healthy controls? I noticed that the healthy controls are missing in the previous works of this group considering this topic (as well as axes units when presenting temperature time series).
It would be instructive for the reader to show the ROC curve since you mention it.
Author Response
We thank the reviewer for the comments provided. Please see the attachment with our point-by-point response.

Round 2
Reviewer 1 Report
The manuscript has been improved compared to the previous version.
There are a few more minor comments:
It is necessary to include the references on ApEn and SampEn in section 2.2. and 2.3.
It is not common to write std values in the description of experimental data. The final conclusion whether the data fulfilled the condition of stationarity or not should be mentioned.
Ln 197-200: It would be useful to show statistics of a number of matches in ApEn for all observed time series, N=500. For example, the percentage of estimated probabilities on at least 10 matches (weak criteria) and 100 matches (strong criteria).
Author Response
We are grateful to the reviewer for the comments and suggestions provided. We have reviewed the manuscript in accordance with these comments and a point-by-point response is attached.
